# Apoptosis of intestinal epithelial cells restricts *Clostridium difficile* infection in a model of pseudomembranous colitis

Pedro H.V. Saavedra [1,2], Linyan Huang[1,2,3], Farzaneh Ghazavi[2,4], Stephanie Kourula[2,4], Tom Vanden Berghe[2,4], Nozomi Takahashi[2,4], Peter Vandenabeele [2,4] & Mohamed Lamkanfi[1,2,5]

*Clostridium difficile* is the leading cause of pseudomembranous colitis in hospitalized patients. *C. difficile* enterotoxins TcdA and TcdB promote this inflammatory condition via a cytotoxic response on intestinal epithelial cells (IECs), but the underlying mechanisms are incompletely understood. Additionally, TcdA and TcdB engage the Pyrin inflammasome in macrophages, but whether Pyrin modulates CDI pathophysiology is unknown. Here we show that the Pyrin inflammasome is not functional in IECs and that Pyrin signaling is dispensable for CDI-associated IEC death and for in vivo pathogenesis. Instead, our studies establish that *C. difficile* enterotoxins induce activation of executioner caspases 3/7 via the intrinsic apoptosis pathway, and demonstrate that caspase-3/7-mediated IEC apoptosis is critical for in vivo host defense during early stages of CDI. In conclusion, our findings dismiss a critical role for inflammasomes in CDI pathogenesis, and identify IEC apoptosis as a host defense mechanism that restricts *C. difficile* infection in vivo.

[1] Department of Internal Medicine, Ghent University, Ghent B-9052, Belgium. [2] VIB-UGent Center for Inflammation Research, VIB, Ghent B-9052, Belgium. [3] School of Medical Technology, Xuzhou Medical University, Xuzhou, Jiangsu 221004, China. [4] Department of Biomedical Molecular Biology, Ghent University, Ghent B-9052, Belgium. [5] Janssen Immunosciences, World Without Disease Accelerator, Janssen Pharmaceutica, Pharmaceutical Companies of Johnson & Johnson, Beerse B-2340, Belgium. Correspondence and requests for materials should be addressed to M.L. (email: mlamkanf@its.jnj.com)

Clostridium difficile is a Gram-positive, toxin-producing and spore-forming obligatory anaerobic bacterial pathogen. It most commonly infects patients that are on antibiotics treatment, which licenses C. difficile outgrowth and intestinal colonization by disrupting the gut microbiota[1]. C. difficile infection (CDI) is thought to account for 15–30% of patients suffering from antibiotic-associated diarrhea, and 3–8% of patients with CDI progress to fulminant infection, a potentially life-threatening inflammatory condition that may involve severe ileus, toxic megacolon, colonic perforation with subsequent peritonitis, and septic shock. The pathophysiology of CDI is strictly associated with the production of TcdA and TcdB, two major pathogenic enterotoxins with cytotoxic properties that drive the hallmark symptoms of pseudomembranous colitis[2,3]. TcdA and TcdB both are exotoxins that primarily target intestinal epithelial cells (IECs) in the intestinal tract. Once internalized via endocytic mechanisms, they autoprocess in the acidified endosome, and the amino-terminal cleavage fragments gain access to the cytosol[4]. TcdA and TcdB both are glucosyltransferases that selectively monoglucosylate a conserved threonine residue that is critical for GTP binding by members of the Rho, Rac and Cdc42 small GTPase families[5,6]. TcdA/B-mediated monoglucosylation results in Rho GTPase inactivation, disruption of the cytoskeleton dynamics, cell rounding and intestinal epithelial cell cytotoxicity, and intestinal damage and permeabilization, consequently leading to bowel inflammation and life-threatening diarrhea[1].

Recent reports showed that TcdA/B-mediated inactivation of RhoA GTPases in mouse macrophages and human peripheral blood mononuclear cells (PBMCs) engages the Pyrin inflammasome[7–10]. Inflammasomes comprise a set of large multiprotein complexes that recruit and promote cleavage and activation of the cysteine protease caspase-1. Once active, caspase-1 drives processing of the pro-inflammatory cytokines interleukin (IL)-1β and IL-18 into their active forms[11]. In addition, caspase-1 cleaves gasdermin D (GSDMD), the amino-terminal fragment of which inserts in the plasma membrane to induce a lytic cell death mode termed pyroptosis[12–16]. Other intracellular pattern recognition receptors (PRRs) that engage an inflammasome response include the nucleotide-binding oligomerization domain, leucine rich repeat containing (NLR) family members NLRP1, NLRP3 and NLRC4; and the HIN200 family member AIM2[17]. Although inflammasome activation has been extensively documented to mediate protective as well as detrimental responses during bacterial, viral and fungal infections[18], the physiological role of inflammasome activation in the context of CDI is unknown. Moreover, the molecular mechanisms and pathophysiological roles of C. difficile-induced IEC cytotoxicity are unclear. Previous studies that primarily relied on the use of pharmacological agents in immortalized IEC cell lines variably proposed that TcdA and TcdB may induce IEC cytotoxicity via caspase-dependent or -independent mechanisms[19–26].

Here we took advantage of state-of-the-art ex vivo models of C. difficile-infected and TcdA/B-intoxicated primary IEC organoid cultures to define the mechanisms of C. difficile-induced IEC cytotoxicity. Additionally, we explored the pathophysiological role of IEC killing in mice challenged in vivo with live C. difficile. We found that unlike in myeloid cells, the Pyrin inflammasome is not functional in IECs and that the central pyroptosis effector GSDMD is dispensable for C. difficile toxin-induced IEC killing. We further established that genetic inactivation of the necroptosis effector MLKL nor the apoptotic initiator caspase-8 altered the course of TcdA/B-induced IEC killing. Instead, we identified C. difficile toxin-induced activation of the apoptotic executioner proteases caspases 3 and 7 by the intrinsic apoptosis pathway as a critical mechanism driving IEC

death. Finally, we established that the Pyrin inflammasome is dispensable in vivo for host defense against CDI, whereas selective deletion of caspases 3 and 7 in IECs exacerbated bacterial burdens and clinical parameters of CDI during early stages of infection.

## Results

**Inflammasomes are dispensable for toxin-induced IEC cytotoxicity.** Intoxication of both mouse macrophages and human PBMCs with either TcdA or TcdB was recently shown to engage the Pyrin inflammasome, which induces pyroptosis and release of the matured inflammatory cytokines IL-1β and IL-18[7–10]. However, these toxins primarily target IECs in the context of CDI, but whether the Pyrin inflammasome drives toxin-induced IEC death and plays a pathophysiological role during CDI is not known. To explore effector mechanisms that promote IEC cytotoxicity, cell death induction in primary IEC organoids was monitored by time-lapse imaging following intoxication with TcdA and TcdB. Wild-type intestinal organoids displayed a shrinking and disorganized cellular architecture that was associated with incorporation of the cell-impermeant DNA-intercalating agent propidium iodide (PI) already during initial hours of intoxication, and appeared dead by 16 h post-treatment (Fig. 1a). Similarly, intestinal organoids from $Mefv^{-/-}$ mice (lacking expression of the inflammasome sensor Pyrin) incorporated PI and displayed cell death features with kinetics resembling that of wild-type IEC organoids (Fig. 1a), indicating that Pyrin signaling is dispensable for TcdA-induced cytotoxicity of IECs. We next intoxicated intestinal organoids from $Asc^{-/-}$, $C1^{-/-}C11^{-/-}$ and $Gsdmd^{-/-}$ mice to verify whether TcdA-induced IEC death was routed through inflammasomes other than the Pyrin pathway. Notably, IECs lacking the core inflammasome components ASC, caspase-1/11 or GSDMD all were equally sensitive to TcdA-induced cytotoxicity and died with kinetics resembling that of wild-type IEC organoids (Fig. 1b–d). Importantly, TcdB-induced IEC cytotoxicity was unaltered in intestinal organoids from $Mefv^{-/-}$ mice (Supplementary Fig. 1). IEC death was toxin-induced because mock-treated organoids of all analyzed genotypes failed to undergo cell death during the observed timeframe (Supplementary Fig. 2A-D). Together, these results demonstrate that unlike in monocytes and macrophages, Pyrin inflammasome activation and inflammasome-induced pyroptosis are dispensable for IEC cytotoxicity induced by C. difficile toxins TcdA and TcdB.

To clarify why TcdA and TcdB failed to induce Pyrin-dependent pyroptosis in IECs, we analyzed the transcript levels of Mefv (that encode Pyrin) in a public gene expression profile dataset from BioGPS[27] (Fig. 2a) and by tissue real-time qPCR analysis (Fig. 2b). Consistent with a recent report demonstrating that Mefv expression in the intestinal tract is confined to lamina propria cells and absent from the IEC fraction[28], we detected high Mefv transcript levels in bone marrow and myeloid cells (macrophages, monocytes, dendritic cells and neutrophils) whereas adrenal gland, bladder, heart, kidney, thymus, liver, lung and small and large intestines were virtually devoid of Mefv expression (Fig. 2a, b). Consistent with the low transcript levels in intestinal IECs (Fig. 2a), we also could not detect Pyrin expression at the protein level in cell lysates of intestinal organoids grown in vitro (Fig. 2c). As expected, robust Pyrin expression was detected in BMDM cell lysates (Fig. 2c). Together, these results suggest that Pyrin inflammasome signaling is restricted to cells of the myeloid lineage, contrary to the NLRC4 and NLRP3 inflammasomes that are functional in both myeloid cells and IECs[29–32]. In agreement, TcdA, TcdB and FlaTox (a biochemical ligand of the NAIP5/NLRC4 inflammasome[29,33]) all triggered IEC death in primary organoids (Fig. 2d and Supplementary

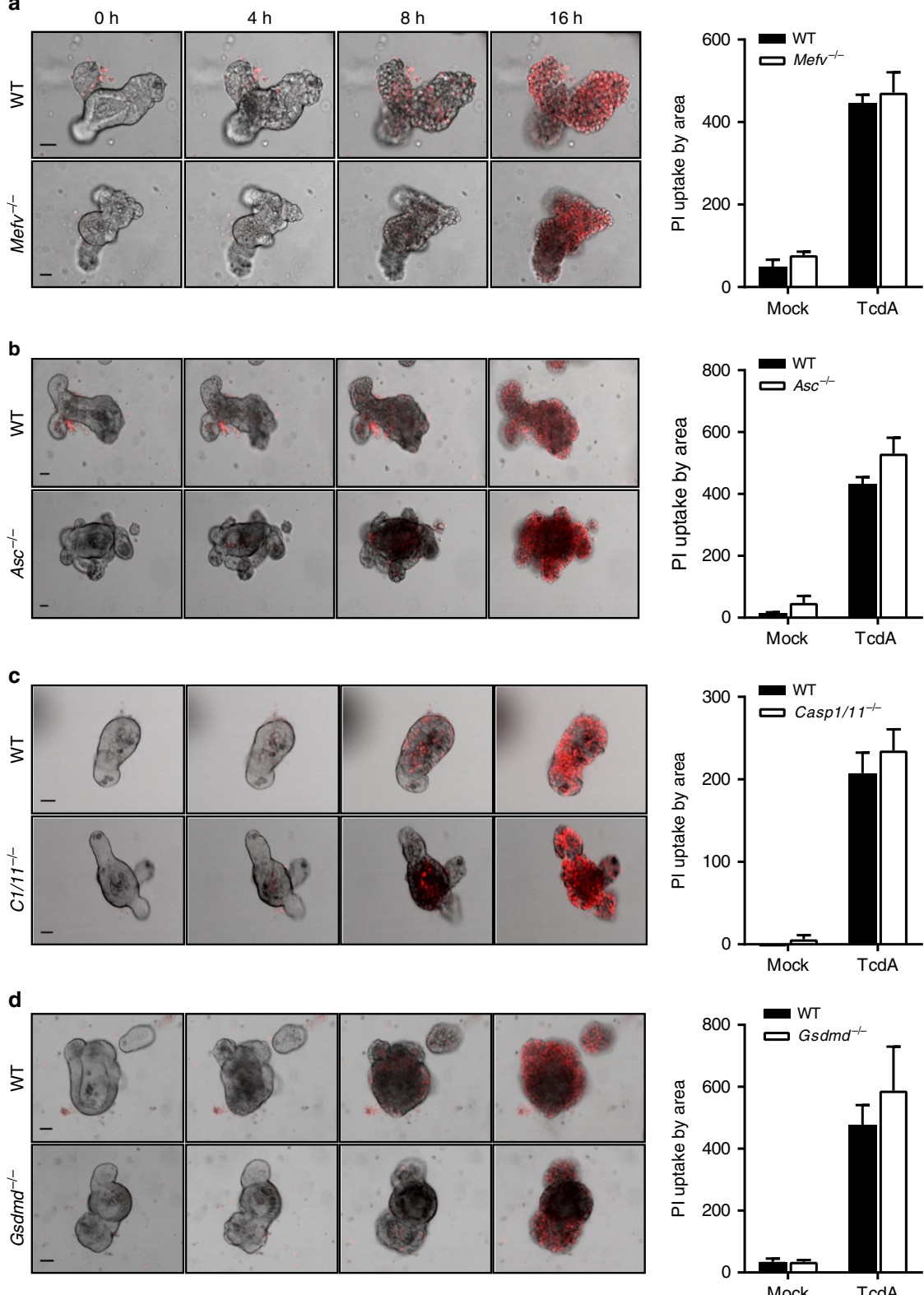

**Fig. 1** Inflammasome activation and pyroptosis are dispensable for *C. difficile* TcdA-induced IEC cytotoxicity. **a–d** Primary intestinal organoids from wild-type (WT) and **a** *Mefv*−/−, **b** *Asc*−/−, **c** *Casp1/11*−/− or **d** *Gsdmd*−/− mice were stimulated with TcdA and PI incorporation analyzed by live-imaging for 16 h. Graphs correspond to PI quantification plotted by organoid area. Scale bars: 30 μm. The data are representative of at least 3 independent experiments

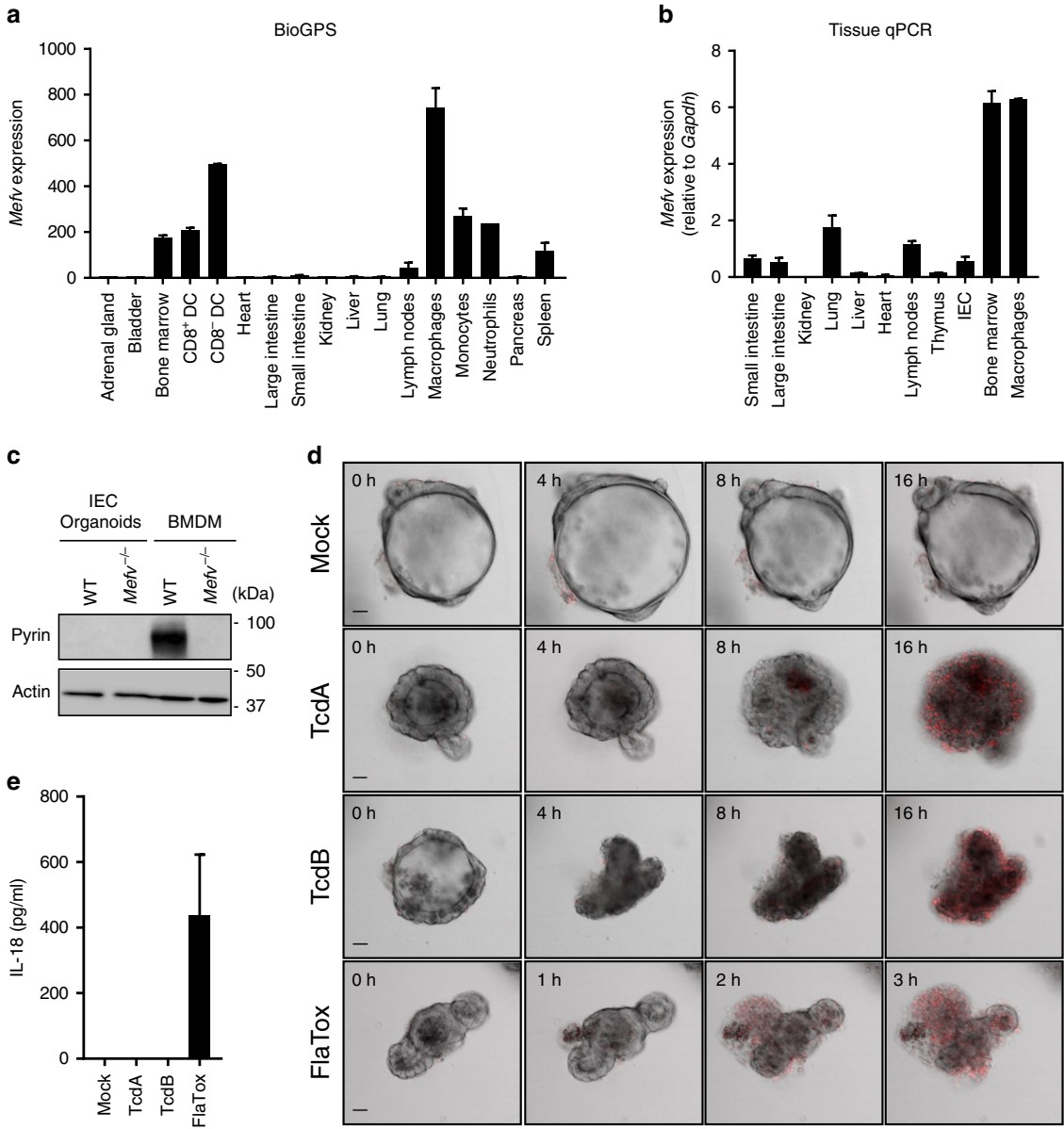

**Fig. 2** Pyrin is not expressed in IECs. **a** Public microarray dataset of *Mefv* expression profile across different cell types and tissues. Expression values relate to fluorescence intensity from Affymetrix chips. **b** Real time qPCR of *Mefv* expression in different tissues and cell types. **c** Cell lysates from primary intestinal organoids of wild-type and *Mefv*$^{--/-}$ mice were prepared and immunoblotted for total Pyrin and β-actin. **d** Primary intestinal organoids wild-type mice were stimulated with TcdA, TcdB or FlaTox and PI incorporation analyzed by live-imaging for 16 h. **e** Primary intestinal organoids wild-type mice were stimulated with TcdA, TcdB or FlaTox for 16 h and culture supernatant was analyzed for IL-18 secretion. Scale bars: 30 μm. Data are representative of 3 independent experiments

Movie 1), but only FlaTox-induced cytotoxicity was associated with release of the inflammasome-dependent cytokine IL-18 (Fig. 2e). Moreover, the morphological changes of IECs undergoing TcdA/TcdB-induced cell death differed dramatically from FlaTox-stimulated IECs undergoing pyroptosis (Supplementary Movie 1), adding further credence to the notion that TcdA/TcdB-induced cytotoxicity differs from pyroptosis. In conclusion, these results demonstrate that Pyrin expression and Pyrin inflammasome signaling likely are restricted to the myeloid compartment, and that *C. difficile* toxins trigger a cell death mode in IECs that differs from inflammasome-induced pyroptosis.

**Caspases 3 and 7 drive *C. difficile*-induced IEC death**. Having ruled out pyroptosis induction as the mechanism of TcdA/B-

induced IEC killing and based on our previous findings (Figs 1 and 2), we next asked whether *C. difficile* toxins elicited an apoptotic response in intoxicated IEC organoids. To this end, lysates of wild-type intestinal organoids that had been challenged with TcdA and TcdB, respectively, were examined by immunoblotting for activation of the apoptotic executioner caspases 3 and 7 (Fig. 3a). We observed robust cleavage of both caspases that was comparable to caspase-3/7 activation levels that were elicited by staurosporine, a broad-spectrum kinase inhibitor and a potent pro-apoptotic chemotherapeutic agent (Fig. 3a). Given these results, we next infected IEC organoids with either a toxicogenic wild-type strain of *C. difficile* or a mutant strain lacking expression of both TcdA and TcdB. Notably, wild-type *C. difficile* infection elicited robust activation of caspases 3 and 7, a response that was blunted in IEC organoids that had been infected with the

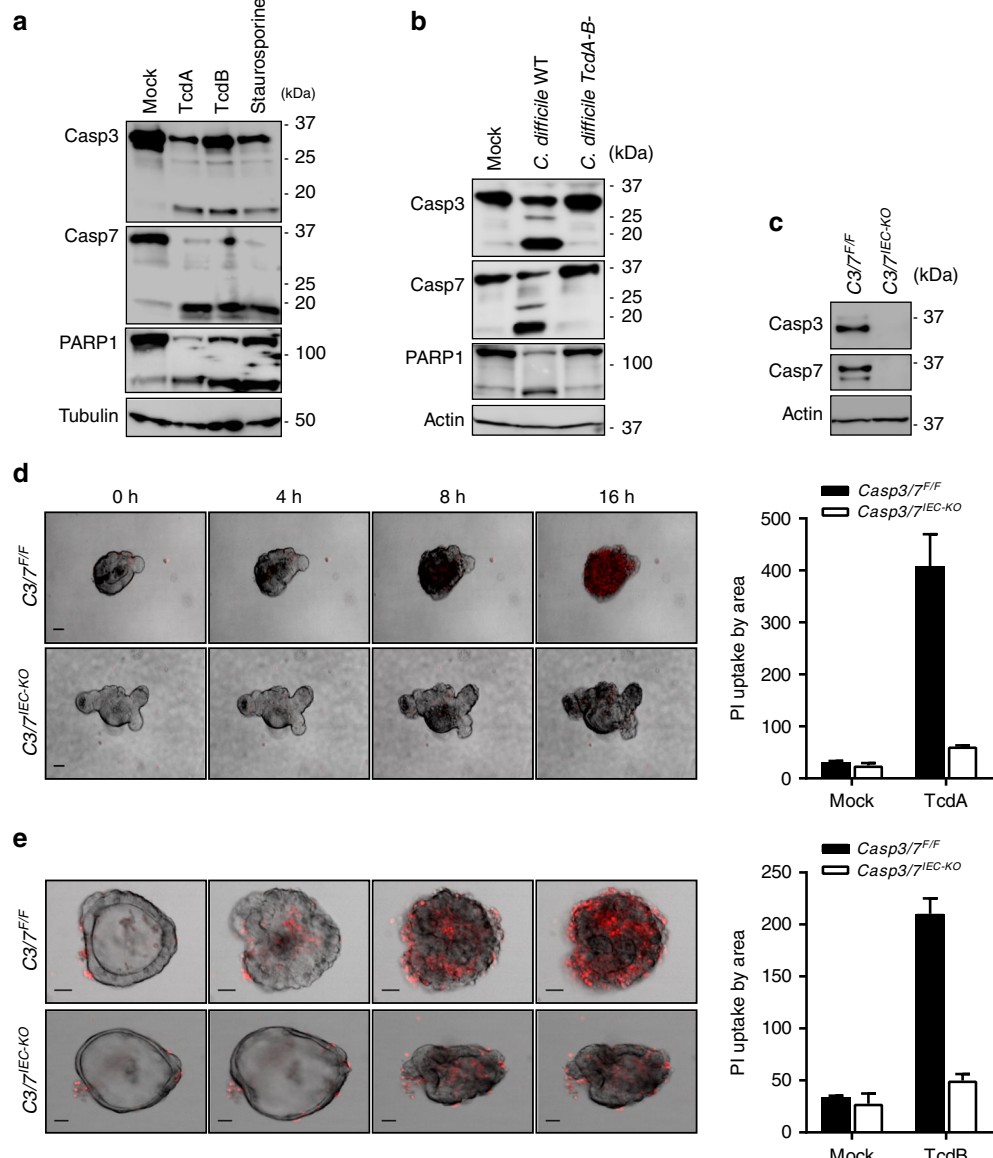

**Fig. 3** Apoptotic executioner caspases 3 and 7 are critical for *C. difficile* toxins-induced IEC death. **a**, **b** Wildtype (WT) primary intestinal organoids were stimulated with **a** TcdA and TcdB for 16 h or staurosporine for 3 h, or **b** infected with a toxigenic and non-toxigenic strain of *C. difficile* for 16 h. Cell lysate was prepared and immunoblotted for caspase-3, caspase-7, PARP and tubulin. **c** Cell lysates from primary intestinal organoids of *Casp3/7^F/F* and *Casp3/7^IEC-KO* mice were prepared and immunoblotted for caspase-3, caspase-7 and β-actin. **d**, **e** Primary intestinal organoids from *Casp3/7^F/F* and *Casp3/7^IEC-KO* mice were stimulated with TcdA (**d**) or TcdB (**e**) and PI incorporation analyzed by live-imaging for 16 h. Graphs correspond to PI quantification plotted by organoid area. Scale bars: 30 μm. The data are representative of 3 independent experiments

toxin-deficient mutant strain (Fig. 3b), in agreement with the established requirement for TcdA/TcdB in *C. difficile*-induced cytotoxicity. Moreover, PARP1 cleavage - a hallmark feature of apoptosis - was readily detected in intestinal organoids that had been intoxicated with TcdA or TcdB (Fig. 3a) or infected with toxigenic *C. difficile* (Fig. 3b), further demonstrating activation of executioner caspases 3 and 7 in these conditions.

To functionally validate the role of caspases 3 and 7 in TcdA/B-induced IEC killing, we produced intestinal organoids from mice with a combined IEC-specific deletion of these executioner caspases (*Casp3/7^IEC-KO*) by breeding mice with conditionally targeted *Casp3* and *Casp7* alleles (*Casp3/7^F/F*) to animals expressing Cre recombinase under control of the IEC-specific Villin promoter (Villin-Cre). Western blot analysis for caspases 3 and 7 showed expression of both executioner caspases in wild-type IECs and confirmed their efficient deletion in *Casp3/7^IEC-KO*

organoids (Fig. 3c). Notably, TcdA and TcdB-induced IEC cytotoxicity was abolished in *Casp3/7^IEC-KO* organoids, which was further supported by a failure to incorporate PI during the studied timeframe (Fig. 3d, e). In marked contrast, TcdA/TcdB-intoxication of control *Casp3/7^F/F* organoids resulted in a swift cell death response within the initial hours of intoxication (Fig. 3d). As expected, mock-treated IEC organoids failed to undergo cell death during the observed timeframe (Supplementary Fig. 3), confirming specificity of these findings. Collectively, these results establish that apoptotic executioner caspases 3 and 7 are critical for *C. difficile*-induced IEC cytotoxicity.

**Bax/Bak pores contribute to TcdA/TcdB-induced IEC killing.** Given that our studies in *Casp3/7^IEC-KO* organoids established a role for apoptotic executioner caspases in *C. difficile* toxin-

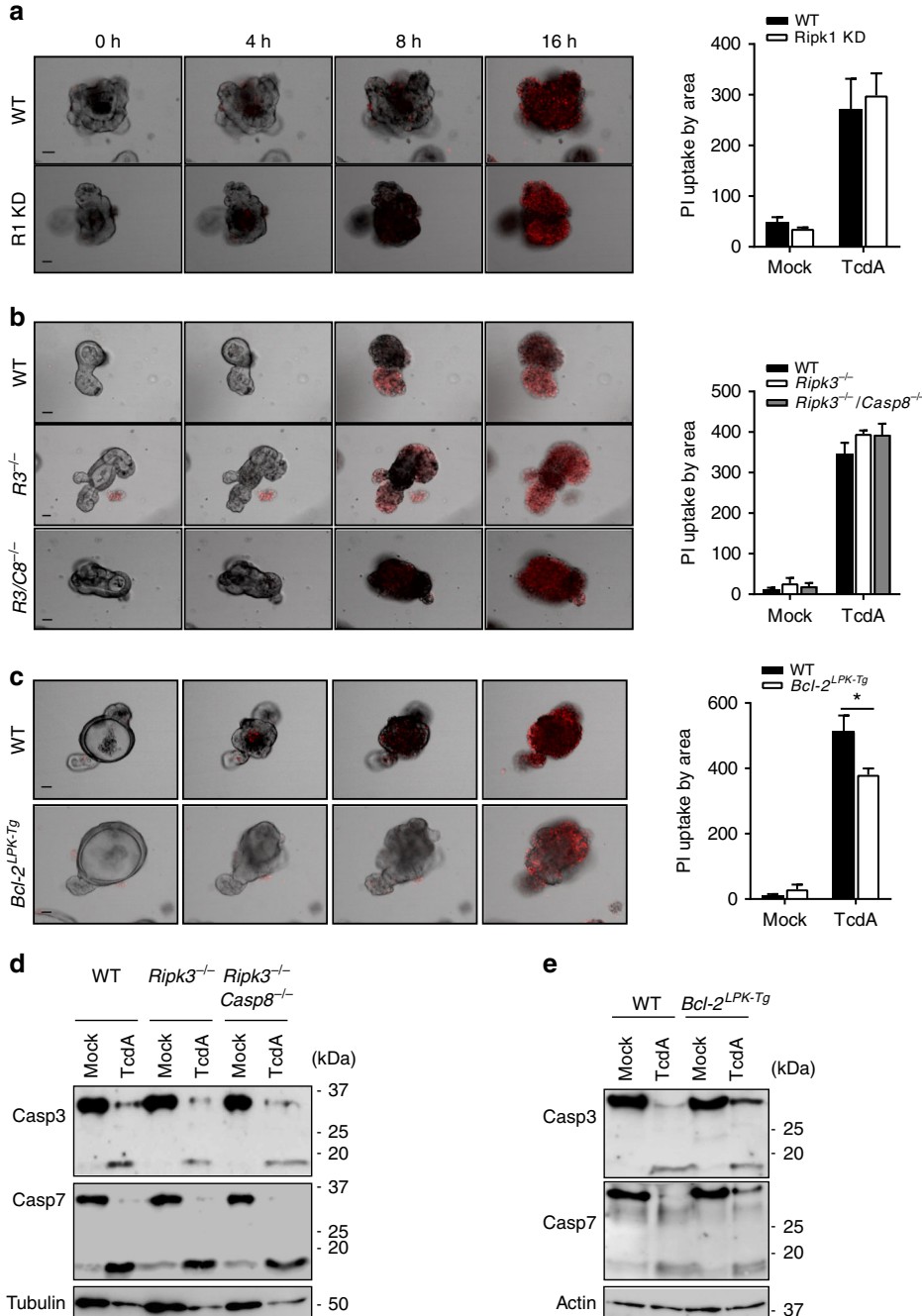

**Fig. 4** RIP kinases and caspase-8 are dispensable, whereas Bax/Bak pores contribute to TcdA-induced IEC killing. **a–c** Primary intestinal organoids from wild-type (WT) and **a** *Ripk1*$^{D138N}$ (RIPK1 kinase dead), **b** *Ripk3*$^{-/-}$ and *Ripk3*$^{-/-}$*Casp8*$^{-/-}$ or **c** *Bcl-2*$^{LPK-Tg}$ mice were stimulated with TcdA and PI incorporation analyzed by live-imaging for 16 h. Graphs correspond to PI quantification plotted by organoid area. Scale bars: 30 μm. **d**, **e** Cell lysates from primary intestinal organoids of **d** *Ripk3*$^{-/-}$ and *Ripk3*$^{-/-}$*Casp8*$^{-/-}$ or **e** *Bcl-2*$^{LPK-Tg}$ stimulated with TcdA were prepared and immunoblotted for caspase-3, caspase-7 and tubulin or β-actin. Data are representative of 3 independent experiments. Data are shown as mean ± SD and were analyzed with 2-way ANOVA. *$P < 0.05$, **$P < 0.01$, and ***$P < 0.001$

induced IEC death, we next sought to analyze the upstream mechanisms promoting activation of caspases 3 and 7. Initiation of classical apoptosis is mediated by two distinct pathways; an intrinsic (mitochondrial) and extrinsic (death receptor) pathway[34]. RIPK1 is a kinase that modulates a regulated necrotic cell death mode termed necroptosis as well as apoptosis induction under certain conditions[35]. However, intestinal organoids from mutant mice that lack RIPK1 kinase activity (R1 KD mice) were killed by TcdA stimulation with similar efficiency as wild-type IECs (Fig. 4a). Moreover, absence of RIPK3—a related kinase that

is essential for induction of necroptosis—failed to protect intestinal organoids against TcdA and TcdB-induced intoxication, regardless whether it was deleted alone or in combination with the apical initiator caspase-8 that drives death receptor-induced apoptosis (Fig. 4b and Supplementary Fig. 4A). Notably, transgenic expression of anti-apoptotic Bcl-2 under control of the LPK promoter for expression in IECs and hepatocytes, supported a slight but reproducible protection against TcdA/TcdB-induced cell death (Fig. 4c, Supplementary Fig. 5A and Supplementary Fig. 6). Intestinal organoid cell death was toxin-induced as

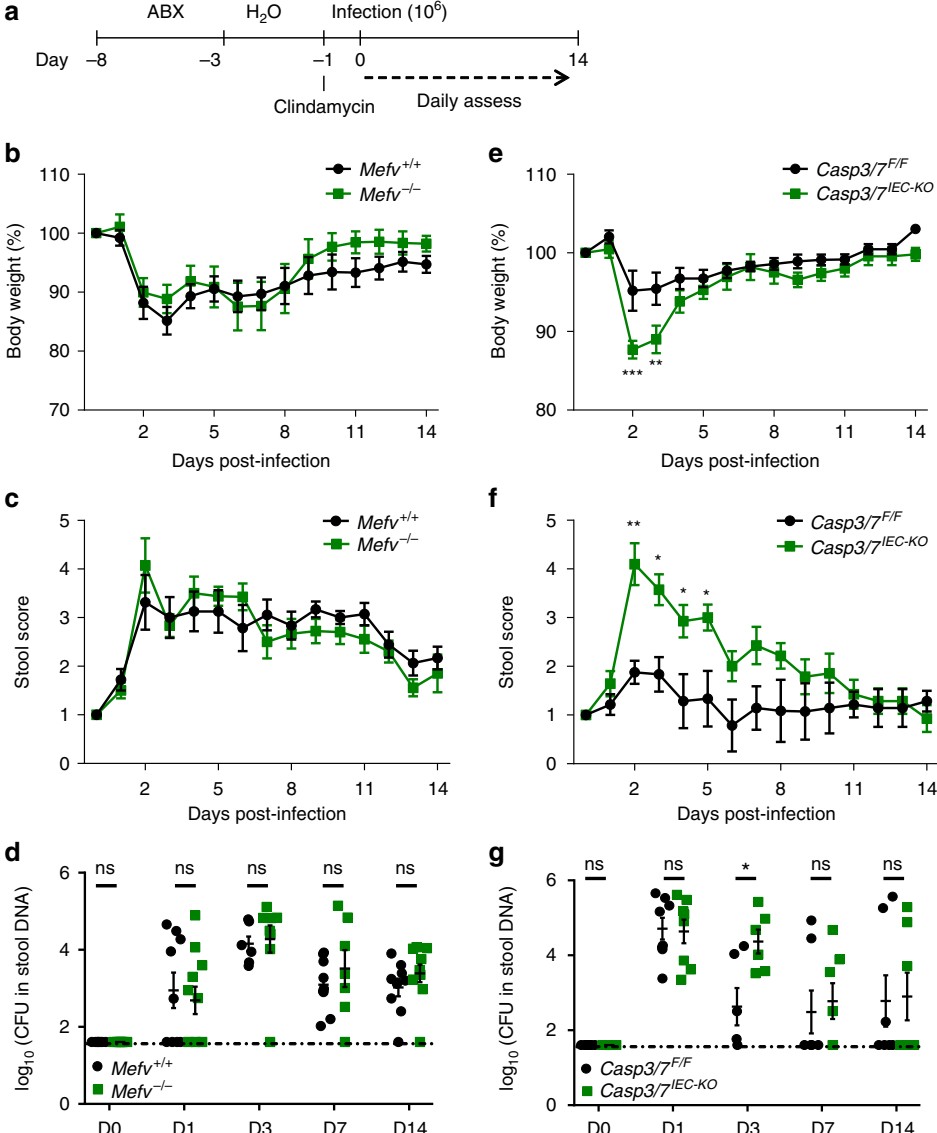

**Fig. 5** IEC apoptosis protects mice during in vivo *C. difficile* infection. **a** Scheme of *C. difficile* infection protocol. **b–d** *Mefv*$^{+/+}$ (*n* = 9) and *Mefv*$^{-/-}$ (*n* = 11) littermates mice were infected with *C. difficile* and monitored for body weight (**b**), stool score (**c**) and bacterial burden in the stool (**d**). **e–g** *Casp3/7*$^{F/F}$ (*n* = 7) and *Casp3/7*$^{IEC-KO}$ (*n* = 7) littermates mice were infected with *C. difficile* and monitored for body weight (**e**), stool score (**f**) and bacterial burden in the stool (**g**). Nonparametric Mann–Whitney *U* test was used to analyze the data. Error bars represent SD. \**P* < 0.05, \*\**P* < 0.01, and \*\*\**P* < 0.001

evidenced by the lack of cytotoxicity in mock-treated controls of all analyzed genotypes (Supplementary Fig. 7A–C). These results suggest that RIP kinases and death receptor-induced apoptosis are dispensable, whereas mitochondrial Bax/Bak pores partially contribute to TcdA/TcdB-induced caspase-3/7 activation and IEC apoptosis. In agreement, TcdA and TcdB intoxication of *Ripk3*$^{-/-}$ and *Ripk3*$^{-/-}$*Casp8*$^{-/-}$ organoids failed to alter cleavage of caspases 3 and 7 (Fig. 4d and Supplementary Fig. 4B). Although activation of caspases 3 and 7 was not substantially altered in Bcl-2 transgenic IECs following TcdA intoxication (Fig. 4e), it was mildly reduced after TcdB intoxication (Supplementary Fig. 5B). In conclusion, these results exclude a role for death receptor-induced apoptosis and necroptosis signaling in *C. difficile* toxin-induced IEC cytotoxicity, and suggest that mechanisms redundant to the pro-apoptotic Bax/Bak pore that lead to mitochondrial outer membrane permeabilization (MOMP) may promote TcdA/B-induced caspase-3/7 activation and IEC apoptosis through the intrinsic apoptosis pathway.

**IEC apoptosis restricts *C. difficile* growth in vivo.** Having established that *C. difficile*-induced IEC death corresponds to caspase-3/7-mediated apoptosis via the intrinsic pathway, we next sought to investigate its pathophysiological role in an established mouse model of live CDI[36]. In this in vivo infection model, littermate mice of the indicated genotypes are treated with a cocktail of antibiotics during 5 days prior to infection in order to create a niche in the intestinal tract that supports *C. difficile* growth (Fig. 5a). Following infection, mice are assessed daily for a number of clinical parameters during a period of 14 days and stool samples are taken at regular intervals to assess bacterial burden. Corroborating our findings that the Pyrin inflammasome is not functional in IECs (Fig. 1), *Mefv*$^{+/+}$ and *Mefv*$^{-/-}$ littermate mice responded to *C. difficile* infection with a comparable decline in body weight over the course of infection (Fig. 5b). Moreover, stool consistency - which is a robust surrogate marker of CDI severity - was equally affected in infected *Mefv*$^{+/+}$ and *Mefv*$^{-/-}$ littermate mice (Fig. 5c). Consistently, longitudinal

quantitative assessment of *C. difficile* replication in stool samples of infected mice confirmed that systemic disruption of Pyrin inflammasome activation failed to impact *C. difficile* replication rates during infection as evidenced by the similar bacterial loads detected in the stool of $Mefv^{+/+}$ and $Mefv^{-/-}$ littermate mice (Fig. 5d).

In marked contrast to Pyrin-deficient mice, selective deletion of executioner caspases 3 and 7 in IECs caused $Casp3/7^{IEC-KO}$ mice to lose significantly more body weight during initial stages of infection relative to littermate $Casp3/7^{F/F}$ mice that do express these executioner caspases in IECs (Fig. 5e). Moreover, stool scores of $Casp3/7^{IEC-KO}$ mice were significantly elevated relative to those of $Casp3/7^{F/F}$ control mice (Fig. 5f). We extended these results by quantifying *C. difficile* burdens in the stool. This analysis showed that $Casp3/7^{IEC-KO}$ and littermate $Casp3/7^{F/F}$ mice had equal *C. difficile* counts on the first day of infection. However, $Casp3/7^{IEC-KO}$ mice continued to present with high infection rates by day 3 post-infection, contrary to $Casp3/7^{F/F}$ mice that started to clear the infection (Fig. 5g). $Casp3/7^{IEC-KO}$ mice eventually controlled *C. difficile* burdens and progressed towards self-limiting disease as evidenced by the reduced bacterial counts detected by day 7 post-infection (Fig. 5g). Collectivity, these results demonstrate that the Pyrin inflammasome is dispensable, whereas caspase-3/7-mediated IEC apoptosis is a critical host defense mechanism that protects against *C. difficile* infection during early stages of CDI.

## Discussion

CDI is a major public health concern. It is estimated that the number of people infected with *C. difficile* is higher than those infected with *Salmonella* species in Europe and the United States[1]. The high incidence of CDI is associated with the emergence of hypervirulent *C. difficile* strains in Europe and North America in the new millennium, which was recently suggested to be associated with the widespread adoption of trehalose as an additive in human diets[37]. Therefore, molecular analysis of *C. difficile*-host interactions and better understanding of how this bacterial pathogen instigates host pathology may improve disease prevention and inform novel therapeutic approaches.

The immune response to *C. difficile* may be both protective and detrimental to the host, depending on its magnitude. Neutrophil recruitment into the intestinal tract consequent to NF-κB signaling via the TLR4 and Nod1 signaling axes is essential for protecting the host against *C. difficile*, as evidenced by decreased survival rates in mice lacking these PRRs[38,39]. Notably, a polymorphism in the human *IL8* gene that results in increased production of IL-8, a neutrophil chemoattractant, is associated with increased risk for *C. difficile* recurrence[40,41], highlighting the importance of a balanced immune response during infection. TLR5 engagement has also been linked to protection against CDI[42]. In addition, innate lymphoid cells (ILCs), mainly ILC1 and to a lesser extent ILC3, also contribute to protection against acute CDI via the induction of IL-22 and IFN-γ[43].

Although inflammasomes are central conduits of innate immune responses, their role in CDI is incompletely understood. TcdA and TcdB were recently shown to engage the Pyrin inflammasome in mouse and human myeloid cells, leading to secretion of IL-1β and IL-18 and induction of pyroptosis[7–9]. However, IECs are the primary target cells of *C. difficile* exotoxins in the context of CDI pathophysiology. Previous studies suggested both protective and detrimental roles for the inflammasome adaptor ASC during CDI[44,45], but whether Pyrin inflammasome signaling impacts on *C. difficile*-induced IEC cytotoxicity and modulates the course of CDI pathophysiology in vivo is unknown. Here, we showed in primary IEC organoid systems

that *C. difficile*-induced IEC cytotoxicity is fully mediated by its exotoxins, and that the Pyrin inflammasome (and inflammasome activation in general) is dispensable for TcdA/TcdB-induced IEC killing, which we linked to absent Pyrin expression and functionality in IECs. This markedly contrasts to the situation in myeloid cells, where Pyrin is highly expressed and inflammasome activation elicits pyroptotic cell death in response to TcdA/TcdB-induced RhoA modification[7–9], thus highlighting the cell type specificity of host-pathogen interactions. Moreover, we provided genetic evidence that TcdA and TcdB-induced IEC death critically relied on caspase-3/7-mediated apoptosis through the intrinsic apoptotic pathway, whereas blockade of both caspase-8-mediated death receptor-induced apoptosis and necroptotic signaling failed to protect from TcdA/TcdB-induced IEC cytotoxicity. Notably, transgenic expression of the anti-apoptotic protein Bcl-2 in intestinal organoids did not fully recapitulate the protective phenotype of IECs lacking caspases 3 and 7, suggesting that mechanisms in addition to the Bax/Bak pore may promote MOMP during CDI. Indeed, Bcl2-transgenic IECs continued to engage caspases 3 and 7 in response to TcdA and TcdB. Thus, although Bax/Bak-independent mitochondrial membrane permeabilization has been reported to induce release of cytochrome c into the cytosol[47–49], future work should address the mechanisms by which *C. difficile* toxins cause MOMP and caspase-3/7 activation. Moreover, a recent report showed that TcdA and TcdB also induce cell death of human IEC organoids[46], and future studies should address whether the molecular cytotoxicity mechanisms we have uncovered in the murine system are conserved in humans.

Induction of cell death upon microbial infections is generally a host defense mechanism[50]. Some pathogens have evolved virulence mechanisms that manipulate cell death pathways in order to promote their replication and dissemination in a hostile host environment. For instance, poxviruses and several bacterial pathogens encode effector proteins that interfere with activation of apoptotic caspases and/or inflammasomes, thereby preventing host apoptosis and pyroptosis, respectively[51,52]. However, it is currently unknown whether IEC cytotoxicity in the context of CDI is detrimental or beneficial to the host. We found that mice lacking apoptotic executioner caspases 3 and 7 selectively in IECs were significantly more susceptible to CDI compared to littermate control mice. In agreement, we measured increased bacterial replication rates in the stool of $Casp3/7^{IEC-KO}$ mice. Notably, $Casp3/7^{IEC-KO}$ mice restored control of CDI infection with delayed kinetics, pointing to apoptosis-independent anti-microbial mechanisms that support *C. difficile* clearance. Although it is tempting to speculate that in rare cases of systemic *C. difficile* toxin dissemination, activation of the Pyrin inflammasome in myeloid cells might exert a detrimental role to the host by promoting excessive myeloid cell pyroptosis and inflammatory tissue damage, our results suggest that the Pyrin inflammasome is dispensable for host defense during self-limiting pseudomembranous colitis. Moreover, the lack of Pyrin expression in IECs prevents this cell type to undergo pyroptosis following *C. difficile* toxin exposure, and instead promotes IEC apoptosis as the dominant IEC death mode during CDI in vivo. These findings suggest that *C. difficile*-induced IEC apoptosis is an early host defense mechanism that contributes to bacterial restriction and toxin production, most likely by promoting clearance of infected cells in the intestinal epithelium by professional phagocytes and infiltrating neutrophils.

## Methods

**Mice.** $Mefv^{-/-}$[8], $Asc^{-/-}$[53], $Casp1^{-/-}Casp11^{-/-}$[54], $Gsdmd^{-/-}$[14], $Ripk3^{-/-}$[55], $Ripk3^{-/-}Casp8^{-/-}$[55] and $Ripk1^{D138N55}$ mice have been described. C57BL/6 ES cells (JM8A3.N1) with conditionally targeted casp3 (clone HEPD0716_4_A08) and casp7 (clone EPD0398_5_C04) alleles were obtained from the European Conditional Mouse

 

Mutagenesis (EUCOMM) program (http://www.mousephenotype.org/about-ikmc/eucommtools) and *Casp3/7* conditional mice (*Casp3/7*$^{fl/fl}$) mice were generated at the VIB Transgenic Mouse Core Facility. *Casp3/7*$^{fl/fl}$ were crossed with Villin-Cre mice to generate mice with specific deletion of Caspase-3 and -7 in IECs. B6.Cg-Tg(Pklr-BCL2)BAk/Orl mice overexpressing human Bcl-2 under the LPK promoter (LPK-Bcl-2) were rederived from embryos and purchased from EMMA (EM:05819). Mice were housed in individually ventilated cages and were maintained in specific pathogen-free facilities of Ghent University. Animal studies were approved by the ethics committees on laboratory animal welfare of Ghent University under permission number EC2016-11.

**Intestinal organoid culture and live imaging.** Primary intestinal epithelial organoids were grown as described before[56]. Briefly, the small intestine and colon were flushed and cut into small pieces that were dissociated in PBS containing 2 mM EDTA for 30 min at 4 °C. After extensive washing, the isolated crypts were pelleted and mixed with 25 μl of Matrigel (Corning) and put in a 24-well plate. After polymerization of the Matrigel, complete culture medium containing advanced DMEM/F12 (Gibco) supplemented with B27 supplement (0.02%, Invitrogen), N2 supplement (0.1%, Invitrogen), N-acetylcysteine (0.0025%, Sigma-Aldrich), mouse epidermal growth factor (mEGF; 0.001%, Invitrogen), and conditioned Rspondin and mNoggin medium was added to the wells. Organoids were seeded and imaged in an 8-well chamber (iBidi) for cell death analysis by real time lapse microscopy or in 24-well plates for Western blotting analysis. Cell death was induced with TcdA (1 μg/ml, Enzo Life Sciences), TcdB (1 μg/ml, List Laboratories), staurosporine (1 μM; SelleckChem), FlaTox (anthrax PA (500 ng/ml, Quadratech) and 1 μg/ml LFn-FlaA[57]) or *C. difficile* (1 × 10⁶). Live-cell imaging was performed on an Axio Observer Z1 (Zeiss) equipped with a CSU-X1 spinning-disk head (Yokogawa) and AxioCam MRm (Zeiss), with a EC Plan-Neofluar 10 × dry objective (numerical aperture [NA] 0.30). Images were acquired every 15 min for 20 h. Data analysis and image reconstruction were performed with ImageJ (NIH) by measuring PI intensity and organoid area. PI intensity was divided by organoid area to obtain final values.

**Clostridium difficile infection.** *Clostridium difficile* strains VPI10463 (toxigenic; TcdA⁺TcdB⁺) and VPI11186 (nontoxigenic; TcdA⁻TcdB⁻) were purchased from ATCC. Glycerol stocks were cultured overnight at 37 °C in anaerobic conditions in BHIS enriched medium (37 g/l brain heart infusion—Gibco; 5 g/l yeast extract—Gibco; 0.03% L-cysteine—Sigma; 0.1% sodium taurocholate—Sigma). For *C. difficile* in vivo infection, mice received antibiotics cocktail (2 mg/ml ampicillin, 0.4 mg/ml kanamycin, 0.035 mg/ml gentamicin, 850 U/ml colistin, 0.215 mg/ml metronidazole and 0.045 mg/ml vancomycin) in drinking water for 5 days followed by 2 days in normal drinking water. One day prior infection, mice were injected with a single intraperitoneal injection of the broad-spectrum antibiotic clindamycin (10 mg/kg). On the day of infection, mice were infected with 10⁶ vegetative cells of *C. difficile* through oral gavage. Mice were checked daily for body weight change and stool consistency. Stool samples were collected on days 0, 1, 3, 7 and 14 post-infection for CFU quantification.

**Stool DNA extraction and C. difficile quantification.** Fecal samples from mice on days 0, 1, 3, 7 and 14 post-infection were collected into sterile soil grinding SK28 tubes (432-0140, Bertin Technologies). Stool samples were processed for gDNA extraction using the QIAamp Fast DNA Stool Mini Kit (51604, Qiagen) according to the manufacturer's instructions. Briefly, stool samples were homogenized in InhibitEX buffer by bead-beating using the Precellys®24 (Bertin Technologies). Fecal homogenates were incubated with proteinase K followed by ethanol precipitation. Stool DNA was diluted in ultrapure sterile water. Quantification of *C. difficile* in stool samples was performed by quantitative PCR (qPCR) as previously described[58]. Briefly, qPCR was performed in 30 ng stool DNA samples using LightCycler 480 SYBR Green I Master Mix kit (4707516001, Roche) in a LightCycler 480 real-time PCR machine (Roche). The standard curve was prepared by isolating *C. difficile* DNA and preparing serial dilutions. Primers used to identify the *tcdB* gene were 5′-GAAGGTGGTTCAGGTCATAC-3′ and 5′-CATTTTC-TAAGCTTCTTAAACCTG-3′.

**Gene expression analysis.** *Mefv* expression across several tissues and cell types were obtained from the BioGPS public database (http://ds.biogps.org/?dataset=GSE10246&gene=54483) and plotted separately. For tissue qPCR, selected tissues were collected from wild-type mice and total RNA was extracted with using the RNeasy kit (74104, Qiagen). RNA (1 μg) was used for cDNA synthesis with iScript reagents (1725037, Bio-Rad). Samples were normalized to the housekeeping gene *Gapdh*. Primers used were murine *Gapdh*, 5′-GGTGAAGGTCGGTGTGAACG-3′ and 5′-CTCGCTCCTGGAAGATGGTG-3′; murine *Mefv*, 5′-TCATCTGCTAAACACCCTGGA-3′ and 5′-GGGATCTTAG AGTGGCCCTTC-3′.

**Western blotting.** Organoids in matrigel were mechanically disrupted and samples were denatured in Laemmli buffer, boiled at 95 °C for 10 min, separated by SDS-PAGE and transferred to nitrocellulose membranes. PBS supplemented with 0.05% Tween-20 (v/v) and 3% nonfat dry milk (w/v) was used for blocking and washing of membranes. Immunoblots were incubated overnight with

primary antibodies against full-length and cleaved caspase-3 (9662 S and 9664 S, 1:1000, Cell Signaling), full-length and cleaved caspase-7 (9492 S and 9491 S, 1:1000, Cell Signaling), PARP (9532 S, 1:1000, Cell Signaling), Pyrin (ab195975, 1:1000, Abcam) or tubulin (ab6046, 1:2000, Abcam), followed by HRP-conjugated secondary antibody raised against rabbit (111-035-144, 1:5000, Jackson ImmunoResearch Laboratories). For some loading controls, β-Actin-HRP (sc-47778 HRP, 1:10,000, Santa Cruz) antibody was used. All proteins were detected by enhanced chemiluminescence (Thermo Scientific). Western blot uncropped scans are available in the Supplementary Information.

**Cytokine analysis.** IL-18 levels in cell culture medium were determined by magnetic bead-based multiplex assay using Luminex technology (Bio-Rad) according to the manufacturer's instructions.

**Statistical analysis.** Statistical analysis was performed using GraphPad Prism 7.0 software. Data are shown as mean ± SD. The data were compared with the nonparametric Mann–Whitney $U$ test for in vivo studies and two-way ANOVA for in vitro studies. $P < 0.05$ was considered statistically significant; *$P < 0.05$, **$P < 0.01$, and ***$P < 0.001$.

**Reporting summary.** Further information on experimental design is available in the Nature Research Reporting Summary linked to this paper.

## Data availability

The data that support the findings of this study are available from the corresponding author upon request. A reporting summary for this article is available as a Supplementary Information file.

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

## Acknowledgements

We thank Vishva M. Dixit, Nobuhiko Kayagaki and Kim Newton (Genentech), and Richard Flavell (Yale) for generous supply of mutant mice. We thank the VIB Transgenic Mouse Core Facility and the VIB Bioimaging Core Facility for technical support. P.V. is supported by Flemish grants (EOS MODEL-IDI consortium, G.0C31.14N, G.0C37.14N, G.0E04.16N, G.0C76.18N, and G.0B71.18N), Methusalem (BOF16/MET_V/007), the Foundation against Cancer (FAF-F/2016/865) and VIB. This work was supported by European Research Council Grant 683144 (PyroPop) and the Baillet Latour Medical Research Grant to M.L.

## Authors contribution:

P.H.V.S. and M.L. conceptualized the project; P.H.V.S. and L.H. performed the experiments; F.G., S.K., T.V.B., N.T. and P.V. provided essential reagents; P.H.V.S. and M.L. analyzed the data and wrote the manuscript; M.L. oversaw the project.

## Additional information

**Competing interests:** The authors declare no competing interests.

