## [Peer Review File · Nature Communications]

Reviewers' comments:

Reviewer #1 (Remarks to the Author):

The paper deals with the role played by the *Clostridium difficile* enterotoxins TcdA and TcdB in *C. difficile* infection (CDI). The authors claim the Pyrin inflammasome, engaged by TcdA/B in myeloid cells, is nonessential for CDI-associated death of intestinal epithelial cells (IEC), and also for in vivo intestinal damage and disease progression. They claim that the production of TcdA/B is mandatory to induce activation of executioner caspases 3/7 via the intrinsic apoptosis pathway and that the caspase-3/7-mediated apoptosis of IECs is critical for in vivo host defense during early stages of CDI. Hence, the principal claim appears to be the pinpointing of infected IECs apoptosis as a host defense mechanism that restricts *C. difficile* infection in vivo.

As concerns the novelty of the claims, it is interesting that the restriction of *C. difficile* infection can relay on IECs apoptosis. However, the fact that pyroptosis, a caspase-1-dependent cell death that is usually induced in macrophages and dendritic cells during infection, does not influence IEC toxicity is not surprising knowing the mechanisms engaged by the two toxins in the target cell. Moreover, the ability of TcdA and TcdB to induce the activation of caspases 3 and 7 and mitochondrial outer membrane permeabilization (MOMP) and therefore apoptosis by the intrinsic pathway is not novel since it has been previously reported in a variety of cell lines (Brito et al. 2002; Qa'Dan et al. 2002; Kim et al. 2005a; Carneiro et al. 2006; Matarrese et al. 2007; Nottrott et al. 2007; Gerhard et al. 2008; Matte et al. 2009; Chumbler et al. 2016). None of these works has been quoted in this paper.

Intestinal organoids, which are an excellent tool to study basic biological processes, have been herein used for investigating *C. difficile* toxins activity. There is however no mention of a previous work that employed human intestinal organoids to study persistence and toxin production by *C. difficile* (Leslie et al. *Infection & Immunity* 2015).

The work presents several criticisms that should be addressed. In particular, the authors claimed that both TcdA and TcdB are important for CDI. However, in the majority of their experiments (Figures 1, 2, 4) they report data obtained only by using TcdA and do not mention data on TcdB neither in the text, arguing why they think act the same, nor in the supplementary data, although discussing the effects of both toxins. Data on both toxins are reported only in Fig. 3. This is a serious criticism of the paper that can lead to false interpretations since it is largely recognized that the two toxins, although sharing the same catalytic activity, yet engage pathways to induce epithelial cells cytotoxicity and apoptosis that in part differ (reviewed by Chandrasekaran and Lacy, *FEMS Microbiol Rev* 2017). Also, it should be underlined that the two toxins cause colonic tissue damage by distinct mechanisms (Chumbler et al. 2016).

Furthermore, the authors did not mention which toxins have been used, if purified (and eventually how, also referring to a previous published work) or received as a gift or purchased. Concerning the strain that they have used, VPI10463 is a well-known strain that produce both TcdA and TcdB. Data in literature, however, indicate that in addition to the Rho, Rac and Cdc42, TcdB also target Rap (Just et al., 1995; Chaves-Olarte et al., 1997) and this additional target could influence the effects on cells. Hence, all the experiments should have been performed by testing both toxins.

As concerns Fig. 5, 'body weight' and 'stool score' are aspecific stress markers and the hypothesis that the observed changes in Figures 5E and 5F may be the consequence of a 'stress' cannot be ruled out. The experimental protocol, in fact, lacks important sham control groups for both Casp3/7F/F and Casp3/7IEC-KO mice.

Reviewer #2 (Remarks to the Author):

Clostridium difficile is the leading cause of nosocomial antibiotic-associated diarrhea and pseudomembranous colitis worldwide. It is established that pathogenesis of *C. difficile* is associated with expression of its two clostridial glucosylating toxins, toxin A (TcdA) and toxin B (TcdB), the action of these toxins on the colonic epithelium - their main host cell target - causing inflammation, tissue damage and fluid secretion, which are the hallmark features of the disease. However, the mechanism of intestinal epithelial cytotoxicity during *C. difficile* infection has remained enigmatic.

Saavedra and colleagues establish here the molecular mechanisms underpinning the cytopathic and cytotoxic effects of *Clostridium difficile* infection and *C. difficile* toxins TcdA and TcdB on the intestinal epithelium. Previous experiments using caspase inhibitors have yielded conflicting results with both caspase-dependent and independent mechanisms having been reported for TcdA and TcdB-induced cell killing of immortalized epithelial cell lines (Brito et al.2002; Qa'Dan et al.2002; Matarrese et al.2007; Nottrott et al.2007; Gerhard et al.2008; Matte et al.2009; Chumbler et al.2016).

The authors of the current study go well beyond the state-of-the-art and make use of cutting-edge primary intestinal epithelial (IEC) organoid systems derived from a series of gene-targeted mice lacking critical components of the Pyrin inflammasome/pyroptosis pathway as well as animals deficient in or transgenic for essential factors of the extrinsic and intrinsic apoptosis mechanisms (Caspase-8, Bcl2), and the necroptosis mediators MLKL and RIPK3 to demonstrate the critical role of intrinsic apoptosis as the cytotoxic mechanism that underpins IEC cytotoxicity by *C. difficile* toxins. Moreover, they generated mice with IEC-selective deletion of executioner caspases 3 and 7 to establish the disease relevance of these caspases in intestinal epithelial cells and their key role in host defense against *Clostridium difficile* infection in an elegant *in vivo* mouse model of self-limiting pseudomembranous colitis.

The manuscript is clearly written and the conclusions are supported by a substantial amount of high quality experiments. I have only a few minor comments for improvement:

1. For the WB data in Fig. 3A and Fig. 3B, inclusion of executioner caspase substrate in these panels would be helpful to further demonstrate activation of the monitored caspases (e.g. PARP1).
2. The authors established in Fig. 1 and in Fig. 5 that the Pyrin inflammasome/pyroptosis pathway in IECs and *in vivo* is dispensable for pseudomembranous colitis due to the absent Pyrin/Mefv expression in IECs, whereas high levels of Mefv expression are detected in myeloid cells. It would be helpful to comment in the discussion section whether authors exclude a role for TcdA/B-induced Pyrin inflammasome activation in the myeloid compartment in the pathological context of (rare) systemic dissemination of *C. difficile* toxins, which is associated with severe, fatal disease in animal models.

We sincerely thank the referees for their time and expertise in evaluating our manuscript and for providing valued input and insightful suggestions that have helped to further improve the quality of our manuscript. In the accompanying revised manuscript, we have included a substantial amount of additional data, including a substantial extension of TcdB related experiments that further strengthen and support our conclusions. We believe our revised manuscript addresses all the aforementioned gaps and hope you will now find this version of the manuscript suitable for publication in Nature Communications. Please find below a detailed response to all enquiries that have been raised.

Reviewers' comments:

Reviewer #1:

The paper deals with the role played by the *Clostridium difficile* enterotoxins TcdA and TcdB in *C. difficile* infection (CDI). The authors claim the Pyrin inflammasome, engaged by TcdA/B in myeloid cells, is nonessential for CDI-associated death of intestinal epithelial cells (IEC), and also for in vivo intestinal damage and disease progression. They claim that the production of TcdA/B is mandatory to induce activation of executioner caspases 3/7 via the intrinsic apoptosis pathway and that the caspase-3/7-mediated apoptosis of IECs is critical for in vivo host defense during early stages of CDI. Hence, the principal claim appears to be the pinpointing of infected IECs apoptosis as a host defense mechanism that restricts *C. difficile* infection in vivo.

As concerns the novelty of the claims, it is interesting that the restriction of *C. difficile* infection can rely on IECs apoptosis. However, the fact that pyroptosis, a caspase-1-dependent cell death that is usually induced in macrophages and dendritic cells during infection, does not influence IEC toxicity is not surprising knowing the mechanisms engaged by the two toxins in the target cell. Moreover, the ability of TcdA and TcdB to induce the activation of caspases 3 and 7 and mitochondrial outer membrane permeabilization (MOMP) and therefore apoptosis by the intrinsic pathway is not novel since it has been previously reported in a variety of cell lines (Brito et al. 2002; Qa'Dan et al. 2002; Kim et al. 2005a; Carneiro et al. 2006; Matarrese et al. 2007; Nottrott et al. 2007; Gerhard et al. 2008; Matte et al. 2009; Chumbler et al. 2016). None of these works has been quoted in this paper.

Response: We thank the referee for the concise, yet accurate summary of the salient findings of our study, and for highlighting our principal finding that TcdA/B-induced intestinal epithelial cell death is driven by caspase-3/-dependent apoptosis as a host defense mechanism that restricts *C. difficile* infection in vivo. Moreover, we believe our reliance on technically advanced primary intestinal epithelial cell (IEC) organoids from a series of gene knockout mice represents another salient aspect of our study. In addition, we generated mice with an IEC-restricted deletion of executioner caspases 3 and 7 to establish the disease relevance of these caspases in intestinal epithelial cells, which represents a unique model to address the in vivo role of IEC cell death in pseudomembranous colitis development. These technically advanced approaches have significantly improved our understanding of the pathophysiological mechanisms of *C. difficile* infection in a more physiological setting compared to most previous studies that have primarily relied on the use of pharmacological inhibitors in immortalized IEC cell lines, and which have led to sometimes conflicting results as to the proposed mechanisms of IEC cell death following TcdA/B

intoxication and *C. difficile* infection. We thank the referee for listing the relevant previous studies and we apologize for not citing them earlier. We now cited them in the updated Introduction and Discussion sections and have included a brief discussion of the reasons why we undertook this study.

Intestinal organoids, which are an excellent tool to study basic biological processes, have been herein used for investigating *C. difficile* toxins activity. There is however no mention of a previous work that employed human intestinal organoids to study persistence and toxin production by *C. difficile* (Leslie et al. Infection & Immunity 2015).

Response: We thank the referee for mentioning this interesting study, which we now incorporated and discussed in the Discussion section:

Page 12: *“Moreover, a recent report showed that TcdA and TcdB also induce cell death of human IEC organoids⁴⁶, and future studies should address whether the molecular cytotoxicity mechanisms we have uncovered in the murine system are conserved in humans.”*

The work presents several criticisms that should be addressed. In particular, the authors claimed that both TcdA and TcdB are important for CDI. However, in the majority of their experiments (Figures 1, 2, 4) they report data obtained only by using TcdA and do not mention data on TcdB neither in the text, arguing why they think act the same, nor in the supplementary data, although discussing the effects of both toxins. Data on both toxins are reported only in Fig. 3. This is a serious criticism of the paper that can lead to false interpretations since it is largely recognized that the two toxins, although sharing the same catalytic activity, yet engage pathways to induce epithelial cells cytotoxicity and apoptosis that in part differ (reviewed by Chandrasekaran and Lacy, FEMS Microbiol Rev 2017). Also, it should be underlined that the two toxins cause colonic tissue damage by distinct mechanisms (Chumblor et al. 2016).

Response: We thank the referee for raising this important point. We concur that findings with TcdA cannot be directly translated to TcdB despite the two toxins sharing similar catalytic activity. We have therefore carried out additional experiments with TcdB intoxication in key genotypes presented throughout our study. We conclude from these experiments that akin to TcdA, TcdB induces IEC cell death through caspases 3 and 7 and independently of Pyrin, as we have shown previously for TcdA. We further show that similarly to TcdA, necroptosis regulator RIPK3 and extrinsic apoptosis initiator caspase-8 are dispensable for TcdB-induced IEC cell death, whereas Bcl2-modulated Bax/Bak pores partially contributed to TcdB-induced cell death. Therefore, we conclude that TcdA and TcdB induce apoptosis in IECs through conserved pathways associated with the intrinsic apoptosis pathway. These important new results have been included in the updated main and supplementary figures of the manuscript.

Furthermore, the authors did not mention which toxins have been used, if purified (and eventually how, also referring to a previous published work) or received as a gift of purchased. Concerning the

strain that they have used, VPI10463 is a well-known strain that produce both TcdA and TcdB. Data in literature, however, indicate that in addition to the Rho, Rac and Cdc42, TcdB also target Rap (Just et al., 1995; Chaves-Olarte et al., 1997) and this additional target could influence the effects on cells. Hence, all the experiments should have been performed by testing both toxins.

Response: We apologize for not mentioning the source of toxins. Both TcdA and TcdB are commercially available and have been purchased from Enzo Life Sciences and List Biological Laboratories, respectively. We have now included this information in the updated Methods section.

As concerns Fig. 5, ‘body weight and ‘stool score’ are aspecific stress markers and the hypothesis that the observed changes in Figures 5E and 5F may be the consequence of a ‘stress’ cannot be ruled out. The experimental protocol, in fact, lacks important sham control groups for both Casp3/7F/F and Casp3/7IEC-KO mice.

Response: We respectfully disagree with the reviewer in this regard and note that body weight and diarrhea scores are widely established and commonly used parameters in the *C. difficile* field to evaluate disease severity upon infection (Etienne-Mesmin et al. 2018 Gut; Cowardin et al. 2016 Nat Microbiol; Abt et al. 2015 Cell Host & Microbe; Buffie et al. 2014 Nature; Chen et al. 2008 Gastroenterology). Additionally, we ran sham controls in our in vivo analyses, and our data - see figure below - showed that the control cohorts did not lose body weight or develop diarrhea contrary to infected mice, demonstrating that the observed change in the aforementioned parameters are not caused by generalized stress, but are infection-induced.

Figure legend. Casp3/7^{F/F} and Casp3/7^{IEC-KO} littermates mice were pre-treated with a cocktail of antibiotics before being orally infected with *C. difficile* or sham infected (PBS), and monitored for body weight and stool score.

Reviewer #2:

Clostridium difficile is the leading cause of nosocomial antibiotic-associated diarrhea and pseudomembranous colitis worldwide. It is established that pathogenesis of *C. difficile* is associated with expression of its two clostridial glucosylating toxins, toxin A (TcdA) and toxin B (TcdB), the action of these toxins on the colonic epithelium - their main host cell target - causing inflammation, tissue damage and fluid secretion, which are the hallmark features of the disease. However, the mechanism of intestinal epithelial cytotoxicity during *C. difficile* infection has remained enigmatic.

Saavedra and colleagues establish here the molecular mechanisms underpinning the cytopathic and cytotoxic effects of *Clostridium difficile* infection and *C. difficile* toxins TcdA and TcdB on the intestinal epithelium. Previous experiments using caspase inhibitors have yielded conflicting results with both caspase-dependent and independent mechanisms having been reported for TcdA and TcdB-induced cell killing of immortalized epithelial cell lines (Brito et al.2002; Qa'Dan et al.2002; Matarrese et al.2007; Nottrott et al.2007; Gerhard et al.2008; Matte et al.2009; Chumbler et al.2016).

The authors of the current study go well beyond the state-of-the-art and make use of cutting-edge primary intestinal epithelial (IEC) organoid systems derived from a series of gene-targeted mice lacking critical components of the Pyrin inflammasome/pyroptosis pathway as well as animals deficient in or transgenic for essential factors of the extrinsic and intrinsic apoptosis mechanisms (Caspase-8, Bcl2), and the necroptosis mediators MLKL and RIPK3 to demonstrate the critical role of intrinsic apoptosis as the cytotoxic mechanism that underpins IEC cytotoxicity by *C. difficile* toxins. Moreover, they generated mice with IEC-selective deletion of executioner caspases 3 and 7 to establish the disease relevance of these caspases in intestinal epithelial cells and their key role in host defense against *Clostridium difficile* infection in an elegant in vivo mouse model of self-limiting pseudomembranous colitis. The manuscript is clearly written and the conclusions are supported by a substantial amount of high quality experiments.

Response: We sincerely thank the referee's appreciation of the quality and novelty of our work. We concur that one of the strengths of our study lays in the use of cutting-edge primary intestinal epithelial (IEC) organoid systems derived from gene-targeted mice, which allowed us to dissect *C. difficile*-induced IEC cell death mechanisms in a more definitive way and in a more physiologically relevant experimental setup of primary organoids than was previously possible with pharmacological approaches in immortalized cell lines. We also appreciate the reviewer acknowledging the elegance of our in vivo mouse model of self-limiting pseudomembranous colitis in mice with an IEC-restricted null allele for executioner caspases 3 and 7. We believe the presented findings are of broad biomedical relevance to microbiology and immunology experts with an interest in *C. difficile* pathogenesis, host-pathogen interactions, innate immune and inflammasome biology and cell death research.

I have only a few minor comments for improvement:

1. For the WB data in Fig. 3A and Fig. 3B, inclusion of executioner caspase substrate in these panels would be helpful to further demonstrate activation of the monitored caspases (e.g. PARP1).

Response: Following the referee's suggestion, we now show that in both settings of Fig. 3A and 3B, PARP1 is cleaved in organoids that have been intoxicated with TcdA/B or infected with *C. difficile*, further confirming that caspases 3 and 7 are active.

2. The authors established in Fig. 1 and in Fig. 5 that the Pyrin inflammasome/pyroptosis pathway in IECs and in vivo is dispensable for pseudomembranous colitis due to the absent Pyrin/Mefv expression in IECs, whereas high levels of Mefv expression are detected in myeloid cells. It would be helpful to comment in the discussion section whether authors exclude a role for TcdA/B-induced Pyrin inflammasome activation in the myeloid compartment in the pathological context of (rare) systemic dissemination of *C. difficile* toxins, which is associated with severe, fatal disease in animal models.

Response: The referee raises an interesting point. While our studies dismissed a role for the Pyrin inflammasome/pyroptosis in self-limiting pseudomembranous colitis, we agree that Pyrin inflammasome activation in the myeloid compartment might be engaged under conditions of the rare, but often fatal systemic dissemination of *C. difficile* toxins. We included a few sentences in the Discussion section to highlight this interesting possibility that may be analogous to the detrimental role of inflammasome activation in various experimental mouse models of septic shock.

Page 13: "Although it is tempting to speculate that in rare cases of systemic *C. difficile* toxin dissemination, activation of the Pyrin inflammasome in myeloid cells might exert a detrimental role to the host by promoting excessive myeloid cell pyroptosis and inflammatory tissue damage, our results suggest that the Pyrin inflammasome is dispensable for host defense during self-limiting pseudomembranous colitis."

REVIEWERS' COMMENTS:

Reviewer #1 (Remarks to the Author):

The Authors have fulfilled all the requests and added new convincing data on TcdB.

I found the revised article acceptable in the present form.

Reviewer #2 (Remarks to the Author):

The authors have addressed all the comments raised by the reviewers.

Reviewers' comments:

Reviewer #1 (Remarks to the Author):

The Authors have fulfilled all the requests and added new convincing data on TcdB.

I found the revised article acceptable in the present form.

Reviewer #2 (Remarks to the Author):

The authors have addressed all the comments raised by the reviewers.

All reviewers indicated that all their comments have been addressed appropriately and supported publication of the revised manuscript.